# Recovery of tropical marine benthos after a trawl ban demonstrates linkage between abiotic and biotic changes

Zhi Wang [1,2,3], Kenneth M. Y. Leung [4,5,7✉], Yik-Hei Sung [6], David Dudgeon [4] & Jian-Wen Qiu [1,2✉]

Bottom trawling, which is highly detrimental to seabed habitats, has been banned in some jurisdictions to mitigate the problems of habitat destruction and overfishing. However, most reports of ecosystem responses to trawling impacts originate from temperate latitudes, focusing on commercial species, and recovery of invertebrate macrobenthos from trawl ban has hardly ever been studied in the tropics. In Hong Kong (lat. 22.4°N), a history of intensive trawling with various types of gears has long degraded coastal ecosystems. To facilitate the recovery of fisheries resources and associated benthic ecosystems, the Government of the Hong Kong Special Administrative Region implemented a territory-wide trawl ban on December 31, 2012. Comparison of surveys conducted in June 2012 (before the trawl ban) and June 2015 (2.5 years after the ban) revealed higher organic contents in sediment and lower suspended-solid loads in water column, as well as a significant increase in site-based abundance, species richness, functional diversity and among-site similarity of macrobenthos after the trawl ban. Our results suggest that the imposition of a trawl ban can be an effective measure for biodiversity conservation in tropical coastal waters.

[1] Department of Biology, Hong Kong Baptist University, Kowloon Tong, Kowloon, Hong Kong, China. [2] Southern Marine Science and Engineering Guangdong Laboratory (Guangzhou), Guangzhou, China. [3] State Key Laboratory of Marine Environmental Science, College of Ocean and Earth Sciences, Xiamen University, Xiamen, China. [4] Division of Ecology and Biodiversity, School of Biological Sciences, The University of Hong Kong, Pokfulam, Hong Kong, China. [5] The Swire Institute of Marine Science, The University of Hong Kong, Cape D'Aguilar, Shek O, Hong Kong, China. [6] Science Unit, Lingnan University, Tuen Mun, Hong Kong, China. [7] Present address: State Key Laboratory of Marine Pollution and Department of Chemistry, City University of Hong Kong, Kowloon, Hong Kong, China. ✉email: kmyleung@cityu.edu.hk; qiujw@hkbu.edu.hk

Bottom trawling, which accounts for roughly 25% of global capture fisheries[1], has increasingly been recognized as a non-sustainable fishing practice[2–5]. It impacts benthic ecosystems in two ways. First, fishing gear disrupts epibenthic sediments, resulting in the loss of habitat complexity and resuspension of sediments into the water column[6–8], reducing the sedimentary organic-matter content[9,10], and increasing turbidity and biochemical oxygen demand in the water column[11,12]. Second, trawling disrupts benthic community structure, selectively removing large-bodied target and non-target species, which are usually K-selected, resulting in a community dominated by relatively small r-selected species[13,14]. Given the significance of these impacts, a number of countries have implemented total or partial bans on bottom trawling within their territorial waters[15,16] or in the international waters they manage[17]. Nevertheless, a recent global review of ecosystem recovery following such bans[8] showed that most ecosystem recovery studies following trawl ban have been conducted in temperate waters, and only one of the 70 studies was conducted in the tropics[18] (i.e. northern Australia). Given the scarcity of information on the recovery of benthic ecosystems for tropical waters[8], empirical studies are urgently needed to determine the rate and time of recovery and the environmental factors that mediate this process.

Modern fishing technologies, including mechanization of fishing vessels and trawling, were introduced into Hong Kong during the 1950s and 1960s[19,20]. By the late 1970s, reductions in total landings as well as catch-per-unit effort, overexploitation of commercially valuable species[21], and destruction of benthic habitats[22,23] had been reported. In consequence, by the mid-1990s, local fisheries authorities realigned their mission from "facilitating production" and "improving productivity" to "sustainable use of fishery resources"[24]. A port survey conducted by the Agriculture, Fisheries and Conservation Department of the Government of the Hong Kong Special Administrative Region (HKSAR)[25] showed that bottom trawling was carried out in almost all Hong Kong's territorial waters, except for a few small protected areas[22,26] and shipping channels. Four hundred trawlers operated partly or wholly within the Hong Kong waters in 2010, accounting for 80% of the total fishing efforts, estimated to be 93% higher than the maximum sustainable fishing yield[27]. To rehabilitate the damaged seabed and depleted fisheries resources, the HKSAR Government implemented a territory-wide ban on all types of trawling activities from December 31, 2012.

Based on surveys collected before (2012)[28] and after (2015) the trawl ban, this study was designed to test three inter-related hypotheses on the abiotic and biotic responses to the trawl ban. Hypothesis 1: total organic matter (TOM) in the sediment would increase and suspended-solid (SUS) loads in the water column would decline because the ban prevents physical disturbance of the seabed by the fishing gear and thus reduces sediment resuspension. Hypothesis 2: the site-based abundance, biomass, species richness, and functional diversity of macrobenthos would increase due to the reduced disturbance and removal by trawling. This rapid recovery of species diversity could be due to the migration of species from previously un-trawled marine parks[22] or from the adjacent waters of Guangdong Province. Hypothesis 3: the niche occupancy of macrobenthos would increase, therefore the problem of habitat fragmentation would be reduced after the trawl ban. Repeated bottom trawling activities could have created highly fragmented benthic habitats due to the potential uneven distribution of trawling efforts, although only gross vessel activities data of the four types of trawlers (i.e., stern, pair, shrimp and hang trawlers) were available in local waters[25]. After the cessation of trawling, the bottom habitats might be less fragmented, which fosters the development of more similar communities in larger spatial zones with higher diversity and abundance. An additional dataset from benthic surveys conducted in 2001[29] was also used as a reference for detecting changes over a longer period of time.

Overall, our study provides the much-needed empirical data to show that a trawl ban is an effective management tool to facilitate rapid recovery of inshore benthic ecosystems (i.e. higher organic contents in sediment and lower suspended-solid loads in water column) and associated biodiversity (i.e. increased abundance, species richness, functional diversity and among-site similarity of macrobenthos) in the tropical coastal waters within a period of 3 years (Fig. 1).

## Results

**Abiotic responses.** Among the 11 abiotic variables tested (Fig. 2b; Supplementary Table 1), four (total organic matter, TOM;

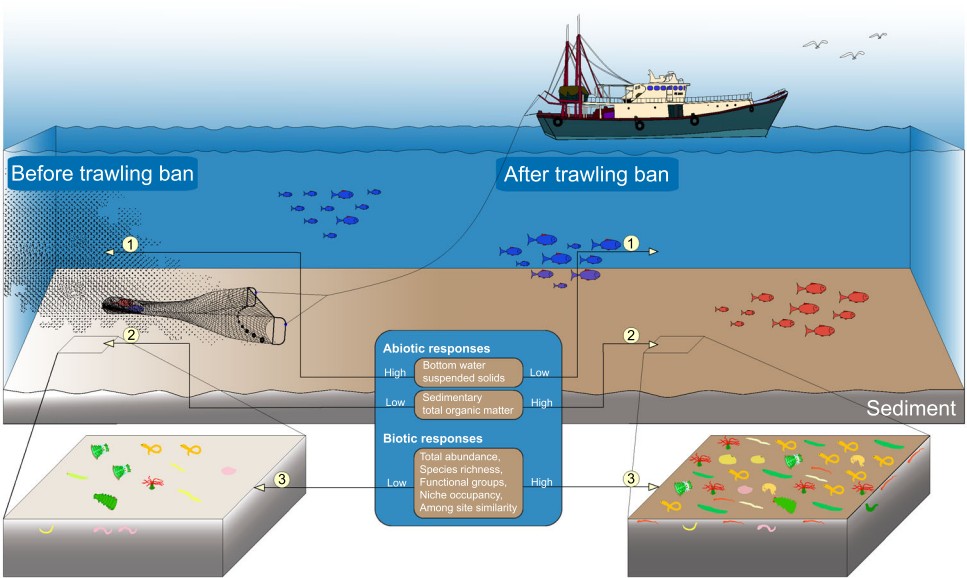

**Fig. 1 An infographic showing the abiotic and biotic responses to the territory-wide trawl ban in Hong Kong waters.** Abiotic responses to the trawl ban include: (1) lower bottom water suspended-solid loads, (2) higher sedimentary organic contents; and Biotic responses include: (3) higher site-based abundance, species richness, functional diversity, niche occupancy and among-site similarity of macrobenthos.

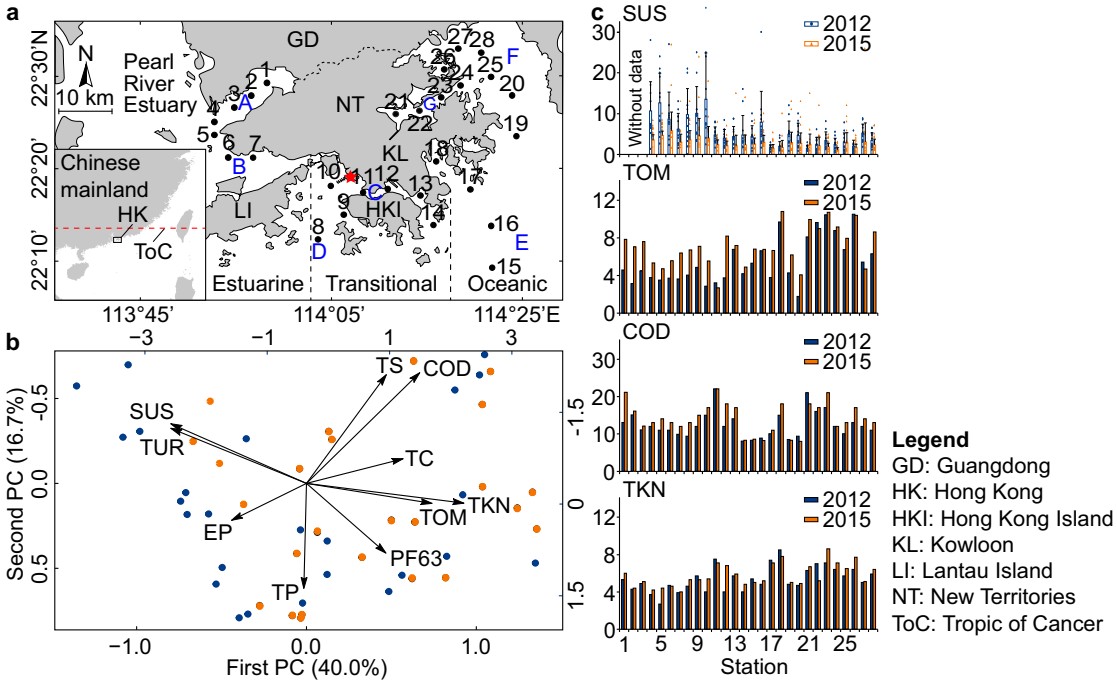

**Fig. 2 Survey sites in Hong Kong waters and comparison of abiotic parameters before (2012) and after (2015) the trawl ban. a** The 28 survey sites located in three hydrological regions (i.e. Estuarine, Transitional and Oceanic) and seven Water Control Zones (i.e. (A) Deep Bay, covering sites 1, 2, 3, 4; (B) Western Waters, covering sites 5, 6, 7; (C) Victoria Harbour, covering sites 9 and 10 outside the harbour, and sites 11, 12, 13, 14 inside the harbour; (D) Southern Waters, covering site 8; (E) Eastern Waters, covering sites 15, 16, 17, 18, 19; (F) Mirs Bay, covering sites 20, 25, 26, 27, 28; (G) Tolo Harbour, covering sites 21, 22, 23, 24) in Hong Kong. The red pentagram represents the location of Stonecutters Island. **b** Principal components analysis (PCA) biplot showing changes in abiotic variables towards increased total organic matter and decreased suspended-solid loads in the survey sites after the trawl ban. Blue and orange circles represent data from the 2012 and 2015 surveys, respectively. Positive or negative correlations between abiotic variables and the two principal components are represented by the direction of the arrows. Abbreviations of abiotic variables refer to Supplementary Table 1. **c** Abiotic variables with significant changes (paired samples *t*-tests) between the 2012 and 2015 datasets. For SUS in **c**, error bars represent the means + standard deviation (SD) of 11 or 12 independent experiments. Between 2012 and 2015, there was a significant decline in bottom water SUS (mg/L), but significant increases in sedimentary TOM (w/w%), COD (1000 mg/kg) and TKN (100 mg/kg).

chemical oxygen demand, COD; total Kjeldahl nitrogen, TKN; and total volatile solids, TVS) significantly increased (paired samples *t*-test, $t = -2.446$ to $-6.101$, $df = 27$, $P = 0.0213$ to $P < 0.001$), while suspended solids (SUS) (paired samples *t*-test, $t = 2.885$, $df = 24$, $P = 0.00815$) and electrochemical potential (paired samples *t*-test, $t = 7.166$, $df = 27$, $P < 0.001$) significantly decreased between 2012 and 2015 (Supplementary Table 2). Compared to the earlier survey, TOM, TVS and COD declined significantly between 2001 and 2012 but increased significantly from 2012 to 2015. However, mean SUS declined significantly between 2001 and 2012, and declined further in 2015 (Supplementary Table 2). Mean reduction of SUS in the latter three years was 0.45 mg/L/yr, substantially higher than the 0.22 mg/L/yr over the 11 years prior to the trawl ban. There was no apparent correlation between monthly precipitation and mean SUS (Pearson correlation coefficient = $-0.062$, $n = 180$, $P = 0.407$). Sites in the semi-enclosed Tolo Harbour (Sites 21, 22, 23 and 24) and Port Shelter (Site 18) had much higher sedimentary TOM, COD, TKN and lower SUS than sites in more open waters (Fig. 2c).

Principal components analysis (PCA) reduced the 10 abiotic factors (exclude TVS with variance inflation factor (VIF) > 10) to two principal components with an eigenvalue >1, which together captured 56.7% of the total variability (Fig. 2b; Supplementary Table 3). Comparing the 2012 and 2015 surveys shows that the loadings of most of sites have increased along the first principal component, reflecting higher sedimentary TOM, finer particle sizes, as well as lower SUS and turbidity (TUR) after the trawl ban (Fig. 2b; Supplementary Table 2).

**Biotic responses**. Although there was a minor decline in the cumulative total number of collected species from 263 (123 families in 8 phyla) in 2012 to 254 (119 families in 8 phyla) in 2015, significant increases in other variables (i.e. total abundance, number of species per site, Margalef's richness index (*d*), functional groups, the abundances of all three motility groups (motile, discretely motile and sessile benthos), and the abundances of three trophic groups (collectors, carnivores and surface deposit feeders)) and a significant decrease in Pielou's evenness index (*J*) were noted (Table 1; Fig. 3a, c, d; Supplementary Fig. 1a–e, g). Total biomass (65.50 and 70.81 g per 0.5 m²) was similar between the two surveys across all sites (Fig. 3b). Among the previously 23 trawled sites, the mean of this parameter generally increased from 17.39 to 50.05 g per 0.5 m² (Supplementary Table 4). The Shannon–Wiener diversity index (*H'*) and the AZTI's Marine Biotic Index (AMBI) did not show significant territory-wide differences between the two surveys, but 19 sites (vs. 9 sites) had higher *H'* values, while 16 sites (vs. 12 sites) had decreased AMBI values after the trawl ban (Supplementary Fig. 1f, h). Spatially, the ratios (2015:2012) of biotic responses varied in previously un-trawled and trawled waters, with higher ratios in the trawled sites for 12 out of 18 biotic responses, while lower ratios for six (i.e. *J*, AMBI, functional diversity, collectors, carnivores and omnivores) (Supplementary Fig. 2). Besides, there were significant changes in 12 out of the 18 biotic variables at previously trawled sites, comparing to un-trawled sites where only one variable showed a significant change (Supplementary Table 4).

**Table 1 Comparison in biotic variables before (2012) and after (2015) the trawl ban in surveyed 28 sites in Hong Kong waters.**

| Variables | 2012 | | 2015 | | t | n | P |
|---|---|---|---|---|---|---|---|
| | Mean | S.D. | Mean | S.D. | | | |
| Total abundance | 253 | 508 | 848 | 1119 | −4.581 | 28 | <0.0001*** |
| Total biomass | 65.50 | 263.68 | 70.81 | 176.15 | −0.087 | 28 | 0.932 |
| Number of species | 27.5 | 18.3 | 48.3 | 26.0 | −6.839 | 28 | <0.0001*** |
| Margalef's richness index, d | 5.33 | 2.85 | 7.30 | 3.51 | −5.179 | 28 | <0.0001*** |
| Pielou's evenness index, J | 0.77 | 0.16 | 0.68 | 0.14 | 3.306 | 28 | 0.003** |
| Shannon–Wiener diversity index, H' | 3.38 | 1.13 | 3.61 | 1.27 | −1.304 | 28 | 0.203 |
| AZTI's Marine Biotic Index, AMBI | 2.53 | 0.80 | 2.30 | 0.78 | 1.161 | 28 | 0.256 |
| Functional diversity | 2.46 | 0.77 | 2.46 | 0.77 | 0.052 | 28 | 0.959 |
| Functional groups | 11.6 | 5.4 | 14.8 | 5.7 | −4.402 | 28 | 0.0002*** |
| Collectors | 111.5 | 326.3 | 285.7 | 375.4 | −2.780 | 28 | 0.0098** |
| Burrowers | 45.7 | 150.8 | 72.7 | 96.4 | −0.816 | 28 | 0.422 |
| Carnivores | 37.0 | 40.3 | 228.5 | 341.1 | −3.238 | 28 | 0.003** |
| Surface deposit feeders | 3.3 | 9.1 | 84.2 | 159.1 | −2.822 | 28 | 0.008** |
| Omnivores | 34.4 | 89.6 | 30.0 | 68.3 | 0.632 | 28 | 0.533 |
| Suspension feeders | 21.7 | 60.3 | 146.4 | 500.3 | −1.319 | 28 | 0.198 |
| Discretely motile | 126.9 | 338.4 | 335.9 | 702.0 | −2.852 | 28 | 0.008** |
| Motile | 118.9 | 276.1 | 489.4 | 714.6 | −4.044 | 28 | 0.0004*** |
| Sessile | 7.6 | 16.9 | 22.2 | 28.6 | −2.943 | 28 | 0.007** |

Asterisks indicate significant differences, i.e.
**$P$ < 0.01,
***$P$ < 0.001.

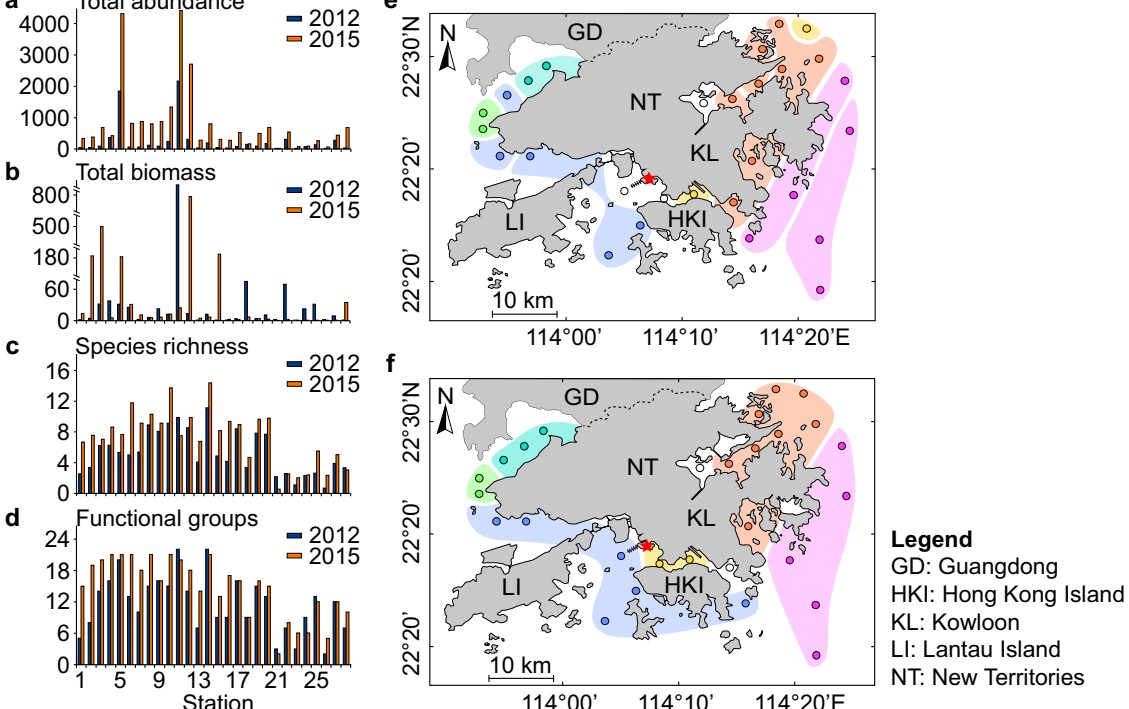

**Fig. 3 The biotic responses and spatial distribution of macrobenthos among-site groups before (2012) and after (2015) the trawl ban. a** Total abundance (number of individuals). **b** Total biomass (g). **c** Species richness. **d** Functional groups. The biotic variables at each site were calculated based on the pooled data from five grab samples covering a total area of 0.5 m$^2$. **e** Site groups in the 2012 survey before the trawl ban. **f** Site groups in the 2015 survey after the trawl ban. The distribution pattern of site groups was made based on the results of CLUSTER analysis (Supplementary Fig. 3) showing nine significantly distinct site groups before and six after the trawl ban (SIMPROF procedure, $P$ < 0.05).

After the trawl ban, there were increases in niche occupancy (referring to the percentage of sites where a certain species was found: 10.45% vs. 18.90%; independent samples $t$-test, $t$ = −5.840, $n1$ = 263, $n2$ = 254, $df$ = 426.3, $P$ < 0.001). The mean Bray–Curtis similarity coefficients ($S_{jk}$) between sites (20.5% vs. 31.3%, paired samples $t$-test, $t$ = 18.49, $n$ = 378, $P$ < 0.001) and the mean intragroup similarity (36.9% vs. 48.9%, independent samples $t$-test, $t$ = 2.388, $n1$ = 9, $n2$ = 6, $df$ = 13, $P$ < 0.05) had

significantly increased, but the number of similar site groups (SIMPROF test, 1000 permutations, $P$ < 0.05; Fig. 3e, f; Supplementary Figs. 3a, b) had decreased from 9 to 6. In other words, the macrobenthic communities had become less fragmented after the trawl ban.

A canonical analysis of principal coordinates (CAP) including three datasets collected before (2001 and 2012) and after (2015) the trawl ban provided us a broader temporal pattern of

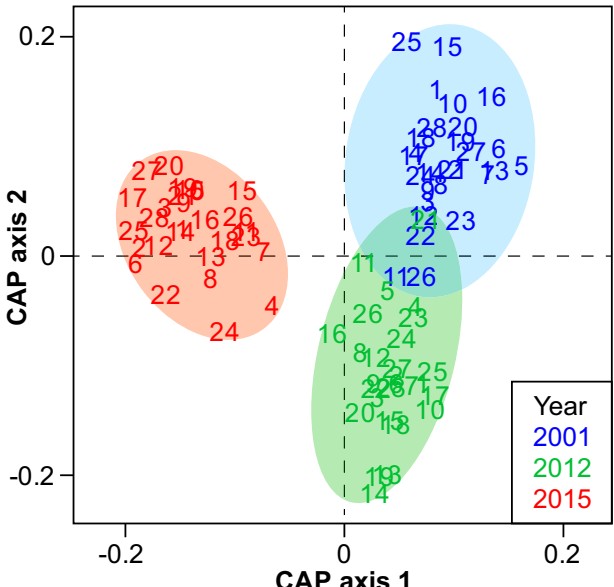

**Fig. 4 Results of Canonical Analysis of Principal Coordinates (CAP) comparing the abundances of macrobenthos in the 28 sites among the three surveys conducted in 2001, 2012 and 2015.** The analysis was based on fourth-root transformed family abundance data (resemblance: Bray–Curtis similarity; number of permutations = 999). Data from the three surveys diverge significantly from each other (pairwise tests, $P < 0.01$).

macrobenthic community changes (Fig. 4). Macrobenthic communities were significantly different between each two of the three surveys (pairwise tests, $P < 0.01$), but the 2015 survey sites were more widely separated from the 2001 and 2012 survey sites along both the axes 1 and 2. Although only 2.5 years had elapsed since the trawl ban, the magnitude of increase in similarity of macrobenthic communities in 2015 was substantially higher than that during the longer period (11 years) before the trawl ban between 2001 and 2012 (Supplementary Fig. 4). The increased similarity in 2015 apparently indicates a recovery of macrobenthic communities consistent with our hypothesis of predicted trawl ban effects.

## Discussion
Bottom trawling can affect the seabed habitat and macrobenthic communities directly and indirectly, but their responses after a trawl ban are not well documented in tropical areas. Before the trawl ban, disturbance of the coastal benthic environments in Hong Kong was extremely high; for example, in Tolo Harbour each square meter could be trawled three times a day[30]. This led to a much lower biomass of long-lived and high-value demersal fishery resources in Hong Kong than in most other inshore areas around the world[31]. Encouragingly, our results demonstrated that 2.5 years after the trawl ban in tropical Hong Kong waters, there was remarkable recovery in benthic habitat as shown by the decreases in bottom water SUS and turbidity, increases in sedimentary TOM, TVS and COD (Hypothesis 1); and recovery in the macrobenthic communities as shown by increases in abundance, species richness and functional diversity (Hypothesis 2), as well as reduced fragmentation of macrobenthic communities (Hypothesis 3).

Macrobenthic communities tend to shift from a diverse fauna to dominance by a few opportunistic species along an increasing gradient of organic pollution[32]. However, in Hong Kong, after the ban on prolonged, intensive trawling activities, the significantly increased sedimentary TOM (Fig. 2c) was coupled with

significant changes in biotic parameters, especially increases in species richness per site (Fig. 3c). This interesting phenomenon might be explained by the relatively smaller increases (mean increase = 1.3%) in TOM resuspended from the seabed sediment into the bottom water; in addition, the increase in sedimentary TOM was also coupled with cessation of intensive physical disturbances from trawling and increase in habitat heterogeneity, which could support higher biodiversity. In other words, our data indicate the trawl ban as the compelling driver for the territory-wide increases in sedimentary TOM and the various biotic changes.

Although factors unrelated to the trawl ban, especially rainfall, could have influenced SUS of bottom waters, we found no significant correlation between monthly precipitation and SUS with a dataset including 180 months in 15 years (2001–2015). The continuing decreasing trend of SUS from 2001 to 2015 might have been attributed to large-scale environmental remediation programmes implemented in Hong Kong waters, i.e. measures aiming to control sewage pollution in the Tolo Harbour, Victoria Harbour and Deep Bay areas[28], and the trawl ban. In general, our results showed that the level of reduction in SUS after the trawl ban was substantially higher than that prior to the trawl ban, which clearly indicates that the trawl ban has played a key role in reducing the bottom water SUS during the survey period.

The responses of macrobenthos to bottom trawling differed among animals of different body sizes[2,3] and functional groups[33,34]. Our results showed that both small polychaetes and larger bivalves, as well as different functional groups had notable increases in abundance after the trawl ban. Sessile benthic species are considered especially vulnerable to trawling disturbances[35,36]. Three common families of sessile polychaetes (Ampharetidae, Terebellidae and Maldanidae) increased in abundance and niche occupancy after the trawl ban, which could be potential biological indicators of the recovery of benthic communities. In a parallel study, our team found that the ban was associated with significant increases in mean body size and in the proportion of large individuals of four species of predatory mantis shrimp[37]. Deposit feeders are usually less affected by bottom trawling than other functional groups due to their shorter lives[38]. Nevertheless, our results revealed that the abundance of surface deposit feeders and collectors, which were mainly small-bodied opportunistic polychaetes and amphipods, also increased substantially after the ban. The changes in macrobenthos we observed, especially in the open waters around Hong Kong, appeared to be mainly due to the cessation of physical disturbances of fishing gear to vulnerable, sessile and long-lived taxa, and recovery of benthic habitats (e.g. increased heterogeneity and TOM) allowing them to support higher species richness, biomass and abundance of macrobenthos[39]. The lower benthic diversity in the more landlocked Tolo Harbour (Sites 21, 22, 23 and 24) has been attributed to the historical discharge of contaminants and seasonal hypoxia in bottom waters[29,40], and the lack of biotic recovery in this area indicated that recovery of benthic community post-trawl ban might have been greatly restricted by hydrology and historical pollution.

Post-trawl ban responses could vary depending on trawling intensity[13]. However, there are no accurate data on trawling intensities in Hong Kong waters. Port surveys conducted by the Agriculture, Fisheries and Conservation Department (AFCD)[25] collected information on fishery production, vessel number and catch value in a uniform grid of 347 areas of Hong Kong waters (Supplementary Fig. 5a–d). Since such surveys showed that fishing vessel numbers of the four types of trawlers were relatively high in the Western, Southern and Eastern Waters and in Mirs Bay, it is not possible to compare estimates of trawling intensities with the extent of abiotic and biotic recovery.

A recent review of recovery from trawling impacts in temperate waters showed that the median biomass recovery time for macrobenthos was 3.6 years for trawling using combined, inseparable fishing gears[8]; the recovery time for abundance depended heavily on the type of fishing gear used: 1.05 year for otter trawl, 4.47 years for beam trawl and 0.18 year for towed dredge. However, most of these studies were conducted to measure recovery after one-off trawling episodes. Because trawling in Hong Kong waters was carried out using mixed fishing gears, and the trawling in most locations was conducted numerous times each year, the time to recovery is expected to be different. Besides, it is difficult to estimate the median recovery time without knowledge of the successional sequences of the benthic community in the surveyed area. Nevertheless, as is the case elsewhere, we expect the recovery in benthic diversity and abundance in Hong Kong to be quicker than that of biomass, given that it may take years for long-lived species such as molluscs, to reach their maximum sizes (Supplementary Fig. 6). This hypothesis can be tested when more time series data are available in the near future.

Overall, our results reveal gratifying signs of recovery in both abiotic and biotic components of the benthic ecosystem after implementation of a trawl ban. The adoption of such a management intervention is recommended as a management measure for rehabilitation of benthic ecosystems in tropical coastal waters.

## Methods

**Study sites and field survey**. Faunal and sediment samples were collected on 5–8 June 2012[28] and on 8, 9, 17, 29, 30 June 2015 from 28 sites covering various areas of Hong Kong waters (Fig. 2a). Data from an ecological survey conducted in June–July 2001[29], which applied a sampling method identical to that the current study, were also used for determination of temporal changes in benthic communities. The sampling sites were located by differential GPS and the water depths were measured by echo sounding from the research vessel. Five sediment samples for faunal analysis and one sediment sample for sediment analysis were collected using a 0.1 m² van Veen grab at each site. The faunal samples were gently rinsed through a 0.5 mm-mesh sieve at sea. Residues retained on the sieve, including macrobenthos, were transferred into labelled plastic bags, fixed in 5% formalin solution in seawater and stained with 1% Rose Bengal. Approximately 400 g sediment at each site were scooped into a plastic bag for sediment analysis, kept on ice on board in a cooler, and transported to the laboratory and frozen at −20 °C in a freezer.

**Sample treatment and data collection**. In the laboratory, macrobenthic samples were rinsed with freshwater, picked up from the sieved residues, transferred to 70% ethanol and later identified to the lowest possible taxonomic level. Abundance was determined by counting only specimens with anterior fragment. Samples were then blotted dry with a paper towel and weighed using an electronic balance (Shimadzu AUW220, Japan).

Because benthic communities are largely structured by sedimentary characteristics and bottom water quality[41], several abiotic variables (as listed in Supplementary Table 1), measured by the Environmental Protection Department (EPD)[42] during its regular sediment and water quality monitoring, were used to assess the relationship between benthic community structure and environmental quality. The data from the time corresponding to the study period (June 2012 and June 2015) were extracted from the monitoring sites that corresponded to our sampling sites. Due to the shallow water depth (1.1–3.8 m), natural turbulences (territorial surface flow, tides) strongly affected the bottom sediment in the inner Deep Bay (Sites 1, 2, and 3), leading to much higher surface suspended-solid (SUS) loads and turbidity than the other surveyed sites; besides, corresponding bottom water SUS and turbidity data were not available for these sites. To determine whether there was a temporal trend of SUS and turbidity, mean values of SUS and turbidity data from 12 months before and on the sampling month (July 2000–June 2001, July 2011–June 2012, July 2014–June 2015; measurement once per month) of the other 25 sites, excluded the three sites in the inner Deep Bay, were used in subsequent data analyses. Monthly precipitation data from 2001 to 2015 were downloaded from the Hong Kong Observatory (https://www.hko.gov.hk/sc/wxinfo/pastwx/mws/mws.htm) and two-tailed Pearson correlation test was performed to explore the correlation between precipitation and SUS using the SPSS Statistics v.17.0 software. Total organic matter (TOM), a potential determinant of benthic community structure that was not measured in the EPD sediment monitoring program, was determined using our sediment samples. Around 20 g freeze-dried sediment from each sample were dried at 100 °C to a constant weight. The content of TOM was calculated as the weight loss after combustion at 500 °C for 8 h in a muffle furnace [GPC 12/65, Carbolite (UK)].

**Statistics and reproducibility**. Statistical analyses of data focused on comparing the abiotic and biotic variables between the two surveys conducted in 2012 and 2015. Additional comparisons with data from the 2001 survey were provided in the supplementary materials (Fig. 4, Supplementary Fig. 4 and Supplementary Table 2). The abiotic variables with significant changes between the two surveys (i.e. TOM, COD and TKN, number of sites = 28, excluding SUS) are shown as single monitoring data in each site in Fig. 2c; while the SUS data (number of sites = 25) are represented as means + standard deviation (SD) of 11 or 12 individual data points. Eighteen biotic variables were determined as listed in Table 1. Total abundance and biomass data of the five benthic samples from each site were pooled (as shown in Fig. 3a, b) to increase the representativeness of the data for univariate and multivariate analyses. All the rest biotic variables in Table 1 were extracted or calculated from the pooled abundance data with a seabed surface cover of 0.5 m² per site. Six feeding groups (i.e. burrowers, carnivores, omnivores, surface deposit feeders, suspension feeders and collectors as those having both suspension and surface deposit feeding guilds) and three motility patterns (i.e. discretely motile, motile and sessile) in macrobenthos were classified and their abundances are shown in Supplementary Fig. 1a–e. The use of these feeding modes and motility patterns followed previous studies[43–48] except that "burrowers" was an additional category. Four biotic variables of each site, i.e., Shannon–Wiener diversity index (H'), species number, Margalef's richness index (d) and Pielou's evenness index (J) were calculated based on species abundance data using PRIMER 6[49] (Fig. 3c; Supplementary Fig. 1f, g). The three characteristics of the functional groups, i.e. (1) feeding mode, (2) motility pattern and (3) morphological structure used in feeding were determined based on several references[43–48], 35 feeding guilds were thereafter established based on the combination of the three characteristics (Supplementary Table 5), and functional diversity in each site was calculated as H' values based on feeding guild data using PRIMER 6. The benthic organisms were assigned to the five ecological groups (i.e. Group I, II, III, IV and V) of macrobenthos[50], which have been shown to respond differently to increasing organic matter in sediment[51], and the AZTI's Marine Biotic Index (AMBI) values were obtained and calculated using the AMBI v.5.0 software[52]. All of the 429 identified macrobenthos in the two surveys were assigned to relevant taxonomic groups and feeding guilds, whereas the ecological groups could only be assigned for 401 species of the macrobenthos (see Supplementary Data 2).

Paired samples t-test was applied to compare each of the abiotic and biotic variables between surveys; and independent samples t-test was applied to compare mean intragroup similarities of macrobenthic groups clustered between surveys. Both tests were performed using the SPSS Statistics v.17.0 software, and P-values of 0.05 or less were considered statistical significance (*$P < 0.05$, **$P < 0.01$, ***$P < 0.001$). For the abiotic variable as shown in Supplementary Table 2, two paired samples t-tests were performed for each variable; due to use of multiple tests, Bonferroni corrections were adopted and only tests with $P < 0.025$ were considered significant (*$P < 0.025$, **$P < 0.005$, ***$P < 0.0005$). Principal Components Analysis (PCA) was applied to identify key variables that could explain the variances in the environmental dataset. The analysis was run using the "FactoMineR" package[53] in the R-Studio 1.0.143[54] software. Before the analysis, multi-collinearity among the abiotic variables was detected with the VIF (variance inflation factor) command in the "CAR" package[55], and variables that had a clear sign of collinearity (i.e. VIF value > 10) were removed from further analysis[56,57]. Due to the lack of bottom water SUS and turbidity data in the inner sites of Deep Bay, PCA analysis was run without including data from the three sites in this area (i.e. Sites 1, 2 and 3) (Fig. 2a). Multivariate analyses of the macrobenthic communities were conducted using the PRIMER 6 software. The "CLUSTER" procedure in the software package was used to determine the spatial and temporal variations in community structure. Abundance data in 'species-sites' matrices in 2012 and 2015 were fourth-root transformed, paired-sites resemblances were calculated using the Bray–Curtis similarity coefficient, and the hierarchical clustering was conducted using the group-average linking method. Significant site groups from clustering were detected using the "SIMPROF" procedure (1000 permutations, 5% significance level). The "Non-metric multidimensional scaling (NMDS)" procedure was used for constructing 3D configurations of sites based on the Bray–Curtis similarities. Pairwise tests under the "PERMANOVA" procedure were used to compare the Bray–Curtis similarities between surveys and to test if the dissimilarities between surveys were significant ($P < 0.05$). The "Canonical Analysis of Principal Coordinates (CAP)" procedure was further used to plot the ordination of sites in different surveys. Source data for plotting these graphs are provided in Supplementary Data 1 and Supplementary Data 2.

**Reporting summary**. Further information on research design is available in the Nature Research Reporting Summary linked to this article.

## Data availability

All datasets supporting the findings of this study are available from the corresponding authors upon request.

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

## Acknowledgements

This research was substantially funded by Research Grants Council of the Government of the Hong Kong Special Administrative Region via a Collaborative Research Fund (project number HKU5/CRF/12 G) to K.M.Y.L. The writing of this paper was partially supported by the Key Special Project for Introduced Talents Team of the Southern Marine Science and Technology Guangdong Laboratory (Guangzhou) (grant numbers GML2019ZD0404, GML2019ZD0409). We thank Ms. Helen Leung and Dr. Yu Sheung Law for their technical support.

## Author contributions

K.M.Y.L. and J.W.Q. initiated and designed this study. Z.W. and J.W.Q. conducted the surveys. Z.W., J.W.Q. and K.M.Y.L. jointly drafted the main manuscript text. J.W.Q. and K.M.Y.L. jointly supervised the work. D.D. and Y.H.S. improved on the methods of data analyses and result interpretation. All authors contributed to the review, revision and preparation of the manuscript.

## Competing interests

The authors declare no competing interests.

## Additional information

 ns license, unless indicated otherwise in a credit line to the material. If material is not included in the article's Creative Commons license and your intended use is not permitted by statutory regulation or exceeds the permitted use, you will need to obtain permission directly from the copyright holder. To view a copy of this license, visit http://creativecommons.org/licenses/by/4.0/.

