## [Peer Review File · Communications Biology]

Reviewers' comments:

Reviewer #1 (Remarks to the Author):

Wang et al, Post-disturbance recovery for Nature Comms

This is an important paper, it addresses one of the most pressing issues in ecology, biodiversity and marine science. There are very few large-scale (space AND time) assessments of recovery from intensive trawling, most countries are not as brave as the HK Government. I would like to see this paper published, but there are a few issues I would like to see attended to before this:

The paper is well written but it does need one last proofread.

I agree that there is little work on trawling bans in the tropics, but some in Australia. The sentence "Most of the studies that have evaluated post ban ecosystem recovery have been conducted in temperate latitudes 18-20" I would delete - these reference do not match the statement 2 are meta-analysis and they are not recovery studies, in fact one of them is a classic example of a paper where the analysis has outweighed the ecological interpretation of results. The third is very small-scale disturbance study. You get to the point in the next sentence.

Hypothesis 3 - a sentence is needed describing how you developed this idea would be useful - does it come from niche theory or meta-community and β -diversity theory?

A sentence in the final paragraph of the Introduction indicating that the ban was effective would help strengthen the paper.

Trawling is not the only factor to impact the seafloor in Hong Kong I am sure (anoxia/hypoxia, contaminants from the region and traveling down the Pearl River are also likely important and this 'multi-stressor' perspective should be considered in the discussion of the results.

The paper would benefit from a clearer ecological interpretation of the results, its rather descriptive. For instance, I would not have expected the benthic community to become more homogeneous post disturbance. Other (temperate) studies show the opposite trend, but perhaps this system was so heavily impacted this is a transitional feature and a longer time series is needed?

Results - Explain variable acronyms on first use. There is a multiple testing problem with the repeated t-tests and the inferences drawn across variables. I would just test the specific hypothesis and use the larger data set in a multivariate analysis.

Recovery rate and recovery time - I think this analysis is inappropriate. This approach comes from papers by fisheries scientists who do not understand community dynamics and focus on highly impacted systems and the recovery of small macrofauna in terms of total abundance - not diversity, not size structure, not ecosystem function. How is carrying capacity defined and ecologically justified? Most of the data used to define ϕ (Extended data Table 8) are very small scale - this is problematic as we expect scale dependence in recovery rates

Reviewer #2 (Remarks to the Author):

This manuscript describes an important and relevant study to assess the potential impacts of bottom trawling in a tropical benthic ecosystem in Hong Kong waters. The study is based on benthic grab surveys conducted before and 2.5 yrs after a trawl ban was put in place and is focused on specific hypotheses on abiotic and biotic responses. The authors report an increase in sediment organic content, decrease in suspended solids, and an increase in macrobenthic biotic abundance and richness, as one might predict from a cessation in use of bottom-impacting gear. The authors also suggest that recovery trajectory (in years) is consistent with other studies.

There are few studies of trawling impacts in tropical and subtropical waters and, therefore, this study represents an important contribution to the scientific literature and a relatively unique contribution to our understanding of potential trawl impacts and recovery trajectories in tropical waters. The data collection & analytic methods seem appropriate. The analyses were focused on comparing abiotic and biotic variables from surveys before and after the trawl ban using paired t-tests. Macrobenthic assemblages were characterized using PCA and independent t-tests. Data from a literature review of tropical studies was used to gauge recovery time.

Some specific comments include:

1. The upfront figure before the abstract is unclear in that it summarizes results in text that is not clearly linked to either before or after the trawl ban. This is a nice infographic - but could be made to make the hypotheses and results more clear.
2. The authors should add in information on the variability across sites within years, as it is clear that the surveys spanned a range of depths and tidal regimes that could drive intra and inter-annual variances. In particular, the variability in suspended solids within and between survey years may be driven by depth, rainfall, or tidal influence more than by trawl abatement.
3. While the authors mention trawl effort at one set of stations, there is no discussion of the relative distribution of trawl effort across the 28 sampling stations or areas that may influence results. Was bottom trawling really conducted at the very shallow (1-3m) sites in Deep Bay?
4. The literature review in EDTable 9 on macrobenthic abundance from other tropical studies was broad, but perhaps overly so? In particular, it may be worthwhile to see if results differ when very small and not-replicated studies are not included (ie. studies with 1 site and <1m² total sampling area).

Reviewer #3 (Remarks to the Author):

This is a very interesting paper that shows the efficacy of the trawling ban and the rapid recovery of sea-bed organisms in the waters in Hong Kong harbour. In some ways it is unfortunate that a short format journal has been chosen as there is much data of interest. But for a more specialist reader the extended data has some great riches. Little work has been done in tropical and sub-tropical areas and therefore this paper will be very influential in shaping policy.

I have no major substantive criticisms. The Figure is a little confusing as the text at the bottom seems to refer to the prior state not the recovered state. Perhaps this needs adjusting or clarifying with an arrow.

In work around the Isle of Man following cessation of scallop dredging communities diverged over time as the homogenizing effect of the fishing gear ceased (work by brand and group). Here it is the other way round - perhaps because the major differences were created by differences in fishing pressure. Would it be worth speculating whether with time divergence among sites might eventually start to occur with hydrodynamic effects on the sediments??

A little more could be made of the interaction between clean-up in the more enclosed waters and fishing cessation.

Minor points

The writing is a little quaint in places and could do with one more edit. Some examples are given below.

I cannot see line numbers so the authors will need to work from the top down.

Introduction

"intensive trawling of various types"

"......demand in the water column"

among site throughout - not between site (between is OK for before and after)

Results

Did you do Bonferroni corrections as you did many comparisons.

Give parameters in full at first mention - putting them in the supplementary material is bad manners and does not make it easy to read.

FIG explain terms in Legend

Results - Abiotic species not specie

NO need to say reported - they have been impacted.

evidence not evidences

No need to say "data analysis of this study" (delete of this study)

Extended data - Figure 6. This is continuous data - therefore the bars should touch (N equals the area under a frequency histogram)

Responses to Comments from Reviewers

Manuscript Number: COMMSBIO-20-1296A

Manuscript Title: Post-disturbance recovery of tropical marine benthos after a trawling ban: Linkage between abiotic and biotic changes

Reviewers' comments:

Reviewer #1 (Remarks to the Author):

This is an important paper, it addresses one of the most pressing issues in ecology, biodiversity and marine science. There are very few large-scale (space AND time) assessments of recovery from intensive trawling, most countries are not as brave as the HK Government. I would like to see this paper published, but there are a few issues I would like to see attended to before this:

Response: We thank the reviewer for the overall positive assessment, and for providing the specific comments that have led to an improvement of our manuscript.

The paper is well written but it does need one last proofread.

Response: We have followed the suggestion to carefully proofread the manuscript after the revision.

I agree that there is little work on trawling bans in the tropics, but some in Australia. The sentence “Most of the studies that have evaluated post ban ecosystem recovery have been conducted in temperate latitudes 18-20” I would delete - these reference do not match the statement 2 are meta-analysis and they are not recovery studies, in fact one of them is a classic example of a paper where the analysis has outweighed the ecological interpretation of results. The third is very small-scale disturbance study. You get to the point in the next sentence.

Response: We have followed the suggestion to delete this sentence “Most of the studies that have evaluated post ban ecosystem recovery have been conducted in temperate latitudes 18-20”.

Hypothesis 3 – a sentence is needed describing how you developed this idea would be useful – does it come from niche theory or meta-community and β -diversity theory?

Response: We have followed the reviewer’s suggestion to add some information about how we developed Hypothesis 3. This came from the niche theory. Repeated bottom trawling activities could have created highly fragmented benthic habitats due to the potential uneven distribution of trawling efforts, although only gross vessel activities data (i.e., not further divided by vessel types) were available in local waters. After the cessation of trawling, the bottom habitats might be less fragmented, which fostering the development of more similar communities with higher diversity and abundance.

A sentence in the final paragraph of the Introduction indicating that the ban was effective would help strengthen the paper.

Response: We have followed the suggestion and added a sentence in the final paragraph of the Introduction in the revised MS to indicate that the trawling ban was effective.

Trawling is not the only factor to impact the seafloor in Hong Kong I am sure anoxia/hypoxia, contaminants from the region and traveling down the Pearl River are also likely important and this ‘multi-stressor’ perspective should be considered in the discussion of the results.

Response: We agree with the reviewer on this account. In the revised MS, we have followed the suggestion to also highlight and discuss the potential effects of hypoxia and contaminants in the enclosed Tolo Harbour and Port Shelter in the Results and Discussion section.

The paper would benefit from a clearer ecological interpretation of the results, its rather descriptive. For instance, I would not have expected the benthic community to become more homogeneous post disturbance. Other (temperate) studies show the opposite trend, but perhaps this system was so heavily impacted this is a transitional feature and a longer time series is needed?

Response: We agree with the comments that the results need clearer ecological interpretation. We interpret the higher homogeneity of the benthic community among the stations as the result of a more uniform environment after the trawling ban. The bottom trawling activities in Hong Kong waters were heavy and likely highly uneven across the seascape, therefore creating more heterogeneous and fragmented seabed environments. The general increases in species diversity, abundance and biomass are consistent with the less habitat fragmentation and higher habitat stability after the trawling ban.

Results - Explain variable acronyms on first use. There is a multiple testing problem with the repeated *t*-tests and the inferences drawn across variables. I would just test the specific hypothesis and use the larger data set in a multivariate analysis.

Response: We have followed the suggestion to explain variable acronyms on the first use, and have revised the significant levels after adopting Bonferroni corrections for the *t*-tests across abiotic variables between the 2001 vs. 2012 datasets, and between the 2012 vs. 2015 datasets.

Recovery rate and recovery time – I think this analysis is inappropriate. This approach comes from papers by fisheries scientists who do not understand community dynamics and focus on highly impacted systems and the recovery of small macrofauna in terms of total abundance – not diversity, not size structure, not ecosystem function. How is carrying capacity defined and ecologically justified? Most of the data used to define ϕ (Extended data Table 8) are very small scale – this is problematic as we expect scale dependence in recovery rates.

Response: Given the lack of knowledge in the benthic community succession in Hong Kong waters, we agree with the reviewer's comment that it was not possible to calculate recovery time for biomass and abundance. This part of the writing has been substantially reduced and revised to point out the need for more time series data to further evaluate the effectiveness of the trawling ban. In fact, we are persuading the Environmental Protection Department of the Government of Hong Kong Special Administrative Region to provide funding to repeat the benthic study in the near future (2021 summer). This will allow for detection of the further recovery of the macrobenthos, especially the long-lived mollusks with the potential to reach high biomass.

Reviewer #2 (Remarks to the Author):

This manuscript describes an important and relevant study to assess the potential impacts of bottom trawling in a tropical benthic ecosystem in Hong Kong waters. The study is based on benthic grab surveys conducted before and 2.5 yrs after a trawl ban was put in place and is focused on specific hypotheses on abiotic and biotic responses. The authors report an increase in sediment organic content, decrease in suspended solids, and an increase in macrobenthic biotic abundance and richness, as one might predict from a cessation in use of bottom-impacting gear. The authors also suggest that recovery trajectory (in years) is consistent with other studies.

There are few studies of trawling impacts in tropical and subtropical waters and, therefore, this study represents an important contribution to the scientific literature and a relatively unique contribution to our understanding of potential trawl impacts and recovery trajectories in tropical waters. The data collection & analytic methods seem appropriate. The

analyses were focused on comparing abiotic and biotic variables from surveys before and after the trawl ban using paired t-tests. Macrobenthic assemblages were characterized using PCA and independent t-tests. Data from a literature review of tropical studies was used to gauge recovery time.

Response: We sincerely thank the reviewer for providing positive overall comments and helpful suggestions.

Some specific comments include:

1. The upfront figure before the abstract is unclear in that it summarizes results in text that is not clearly linked to either before or after the trawl ban. This is a nice infographic - but could be made to make the hypotheses and results more clear.

Response: We thank the reviewer for the advice, and have followed the suggestion to revise the infographic so that it can convey a clear visual impression of the contrast in benthic community before and after the trawl ban.

2. The authors should add in information on the variability across sites within years, as it is clear that the surveys spanned a range of depths and tidal regimes that could drive intra and inter-annual variances. In particular, the variability in suspended solids within and between survey years may be driven by depth, rainfall, or tidal influence more than by trawl abatement.

Response: We agree with the reviewer that a few parameters could affect the benthic community recovery. Accordingly, we have explored whether factors other than the ones analyzed in this study might have confounded our interpretation of the benthic data. In addition to the analysis on the suspended solids (SUS), we have conducted an additional analysis on the relationship between precipitation and SUS, and the results showed no apparent correlation between precipitation and SUS (Pearson Correlation Coefficient = -0.062, $P > 0.05$, $n = 180$) based on a dataset of 180 months in the 15 years (2001-2015) with consideration of irregular seasonal fluctuations in bottom water SUS. Since the benthic infauna live on and inside the sediment of the seabed, previous studies have found that their community structure is mainly structured by sedimentary parameters. A previous study (Shin et al. 2004) showed very little effects of season on local benthic communities. Nevertheless, we standardized the sampling season in summer to avoid the potential seasonal effect. We have also conducted an exploratory analysis on the potential effect of water depth on benthic community changes, but no significant effect was detected.

Cited reference:

Shin, P. K. S., Huang, Z. G. & Wu, R. S. S. An updated baseline of subtropical macrobenthic communities in Hong Kong. *Mar. Pollut. Bull.* 49, 119–141 (2004).

3. While the authors mention trawl effort at one set of stations, there is no discussion of the relative distribution of trawl effort across the 28 sampling stations or areas that may influence results. Was bottom trawling really conducted at the very shallow (1-3m) sites in Deep Bay?

Response: The port surveys conducted by Agriculture, Fisheries and Conservation Department (AFCD) provided information on fishery production, vessel number and catch value in a uniform grid within Hong Kong waters (AFCD Port Survey 2006; **Fig. S1**). These data show that the fishing intensity by the four types of trawlers (i.e., stern, pair, shrimp and hang trawlers) was high throughout Hong Kong waters, especially the Western Waters, Southern Waters, Eastern Waters and Mirs Bay (**Fig. S1**). Although trawling intensity in the shallow Deep Bay area was not as intense as in deeper waters, there were some shrimp and hang trawlers operating in Deep Bay over the past based on the Port Survey results (**Fig. S1c, d**). In the revised MS, we have revised the Discussion section to add information generated from the Port Survey.

Fig. S1 Distribution of fishing operations by (a) stern trawlers, (b) pair trawlers, (c) shrimp trawlers and (d) hang trawlers in Hong Kong waters in 2006 (source: AFCD Port Survey 2006).

4. The literature review in EDTable 9 on macrobenthic abundance from other tropical studies was broad, but perhaps overly so? In particular, it may be worthwhile to see if results differ when very small and not-replicated studies are not included (i.e., studies with 1 site and $<1\text{m}^2$ total sampling area).

Response: We agree with the reviewer's comment on the difficulty in comparing different studies. This section has been substantially restructured and rewritten to emphasize the potential impact of repeated trawling activities on the benthic community in Hong Kong before the trawling ban, and the need for more time to determine the full recovery, especially for the biomass of mollusks that may take many years to reach maximum size. Nevertheless, our study has clearly shown the early signs of community recovery, as measured by a number of biotic parameters especially species diversity, abundance and niche breath.

Reviewer #3 (Remarks to the Author):

This is a very interesting paper that shows the efficacy of the trawling ban and the rapid recovery of sea-bed organisms in the waters in Hong Kong harbour. In some ways it is unfortunate that a short format journal has been chosen as there is much data of interest. But for a more specialist reader the extended data has some great riches. Little work has been done in tropical and sub-tropical areas and therefore this paper will be very influential in shaping policy. I have no major substantive criticisms.

Response: We sincerely thank the reviewer for the very positive and encouraging overall comments.

The Figure is a little confusing as the text at the bottom seems to refer to the prior state not the recovered state. Perhaps this needs adjusting or clarifying with an arrow.

Response: We have followed the suggestion to edit and improve the infographic in order to clearly show the differences in benthic community before and after the trawl ban.

In work around the Isle of Man following cessation of scallop dredging communities diverged over time as the homogenizing effect of the fishing gear ceased (work by brand and group). Here it is the other way round - perhaps because the major differences were created by

differences in fishing pressure. Would it be worth speculating whether with time divergence among sites might eventually start to occur with hydrodynamic effects on the sediments?

Response: We appreciate the interesting and inspiring comments of the reviewer. In fact, in our study in the marine environment of Hong Kong, the 2015 data (after the trawl ban) showed more clearly the structural effects on benthic communities associated with the gradient of water and sediment quality parameters (e.g., sedimentary TOM) than that of the 2012 data (before the trawl ban) (Fig. 2e, f). There might have been some confusion in our expression of the homogeneity in community structure. We actually meant to say that the communities within the larger spatial zones have become less fragmented with increased biodiversity after the trawling ban.

Please see also our responses to the comments of Reviewer #1 in relation to setting the Hypothesis 3 in our study.

A little more could be made of the interaction between clean-up in the more enclosed waters and fishing cessation.

Response: We have followed the reviewer's suggestion to discuss more about the effects of pollution control measures in the revised MS. We have checked the abiotic/biotic data inside the Tolo Harbour, a semi-enclosed bay. However, due to the high levels of sedimentary total organic matter (TOM) and seasonal hypoxia in bottom waters in the Tolo Harbour, the benthic community was characterized by low species richness, low abundance and biomass, and low functional group diversity. This situation did not change and no apparent recovery was noted after the trawling ban.

Minor points

The writing is a little quaint in places and could do with one more edit. Some examples are given below.

Response: We have followed the suggestion to carefully edit the paper during the revision.

I cannot see line numbers so the authors will need to work from the top down.

Response: We have added the line numbers for the whole text.

Introduction

"intensive trawling of various types"

Response: Line 28---We have revised the sentence from "... a history of intensive, various types of trawling ..." to "... intensive trawling with various types of gears ..."

".....demand in the water column"

Response: Line 46---We have revised the sentence from "... demand of the water column" to "... demand in the water column".

among site throughout - not between site (between is OK for before and after)

Response: Lines 37, 85, and 88---We have revised the text and used "among site throughout" instead of "between-site".

Results

Did you do Bonferroni corrections as you did many comparisons.

Response: we have followed the suggestion to revise the statistical results with inclusion of Bonferroni corrections for multiple comparisons.

Give parameters in full at first mention - putting them in the supplementary material is bad manners and does not make it easy to read.

Response: Lines 101-104---We have followed the reviewer's suggestion to include the full names of the parameters at first mention.

FIG explain terms in Legend

Response: Line 118 and 157---We have followed the comments to show full term names in Legend of Fig.1 and Fig.2.

Results - Abiotic species not specie

Response: Line 232---We have revised the word from "specie" to "species".

NO need to say reported - they have been impacted.

Response: Line 249---We have deleted "reported".

evidence not evidences

Response: Line 272---We have corrected this typo.

No need to say "data analysis of this study" (delete of this study)

Response: Line 347---We have deleted "of this study".

Extended data - Figure 6. This is continuous data - therefore the bars should touch (N equals the area under a frequency histogram)

Response: Line 661---We have revised the figure according to the suggestion.

End of Our Responses

REVIEWERS' COMMENTS:

Reviewer #1 (Remarks to the Author):

I think the authors have done a good job of responding to my previous comments and those of the other reviewers.

Reviewer #2 (Remarks to the Author):

I appreciate the effort the authors made to address reviewer comments and I think the manuscript is significantly improved. Specifically, the authors have adequately addressed my initial comments. I have just a few specific comments on the revised version.

(1) The manuscript could use a bit more editing for grammar. Some examples include:

Lines 85-87:"which fostering...."

Line 90: pay attention to being more consistent with verb tense (e.g. "our study provides....that shows trawl ban was..."

(2) For some portions of the results section the inclusion of extensive and repeated statistical results in parentheses makes it difficult to read (e.g. Lines 140-155) - consider a summary table for some of those statistics?

(3) Line 197 -I would suggest changing the phrase "due to the trawl ban" to something more qualified like "likely due" or "consistent with our hypothesis of predicted trawl ban effects". As you mention, there was remediation and other factors not evaluated that could be contributing to the results observed.

Responses to comments from Reviewers

Manuscript Number: COMMSBIO-20-1296A

Revised Manuscript Title (as suggested by Editor): Recovery of tropical marine benthos after a trawl ban demonstrates linkage between abiotic and biotic changes

Response to Editor(s): We sincerely thank the Editor for the good news that our manuscript will be acceptable for publication in *Communications Biology* after a minor revision. Following the suggestion of the Editor, we have also changed the manuscript title from “Post-disturbance recovery of tropical marine benthos after a trawl ban: Linkage between abiotic and biotic changes” to “Recovery of tropical marine benthos after a trawl ban demonstrates linkage between abiotic and biotic changes”. We have also followed the editorial requests for the final version and checked the final submission file checklist to ensure that all necessary files are present with our final submission.

We have carefully proofread the manuscript and revised it based on the comments from the Editor and Reviewer #2. A list of our point-to-point responses to his/her comments is provided below.

REVIEWERS' COMMENTS:

Reviewer #1 (Remarks to the Author):

I think the authors have done a good job of responding to my previous comments and those of the other reviewers.

Response: We sincerely thank the Reviewer #1 for his/her positive comments.

Reviewer #2 (Remarks to the Author):

I appreciate the effort the authors made to address reviewer comments and I think the manuscript is significantly improved. Specifically, the authors have adequately addressed my initial comments. I have just a few specific comments on the revised version.

Response: We gratefully thank the Reviewer #2 for the encouraging comments and providing additional specific comments that would further improve the clarity and quality of our manuscript.

(1) The manuscript could use a bit more editing for grammar. Some examples include:
Lines 85-87:"which fostering...."

Response: We have followed the suggestion to correct the grammar and improve the use of English throughout the revised manuscript. For example, we have edited Lines 85-87 from ...“which fostering”... to ...“which fosters”...

Line 90: pay attention to being more consistent with verb tense (e.g. "our study provides....that shows trawl ban was..."

Response: We have followed the suggestion to check the consistency of the verb tense of sentences throughout the revised manuscript. We found that in our last submitted version,

Line 90 was actually written as: "...our study will provide the much-needed empirical data to show that the trawl ban is an effective management tool...", with a consistent verb tense in this sentence.

(2) For some portions of the results section the inclusion of extensive and repeated statistical results in parentheses makes it difficult to read (e.g. Lines 140-155) - consider a summary table for some of those statistics?

Response: We have followed the suggestion to remove the statistical results from Abiotic Responses (Lines 100-103) and place the information in Supplementary Table 2 in the revised manuscript. We have also moved the statistical results in Biotic Responses (Lines 140-155) to a summary table (Table 1) to make the sentences more readable.

(3) Line 197 -I would suggest changing the phrase "due to the trawl ban" to something more qualified like "likely due" or "consistent with our hypothesis of predicted trawl ban effects". As you mention, there was remediation and other factors not evaluated that could be contributing to the results observed.

Response: We agree with the comments and followed the suggestion to change the phrase "due to the trawl ban" to "consistent with our hypothesis of predicted trawl ban effects".

End of Our Responses